# SARS-CoV-2 Main Protease Dysregulates Hepatic Insulin Signaling and Glucose Uptake: Implications for Post-COVID-19 Diabetogenesis

**DOI:** 10.3390/pathophysiology32030039

**Published:** 2025-08-04

**Authors:** Praise Tatenda Nhau, Mlindeli Gamede, Andile Khathi, Ntethelelo Sibiya

**Affiliations:** 1Pharmacology Division, Faculty of Pharmacy, Rhodes University, Makhanda 6139, South Africa; tatenhau@gmail.com; 2Human Physiology Department, University of Pretoria, Pretoria 0002, South Africa; mlindeli.gamede@up.ac.za; 3School of Laboratory Medicine and Medical Sciences, University of KwaZulu-Natal, Durban 4001, South Africa; khathia@ukzn.ac.za

**Keywords:** diabetes mellitus, insulin, SARS-COV-2, M^pro^, liver, insulin resistance

## Abstract

**Background:** There is growing evidence suggesting that SARS-CoV-2 may contribute to metabolic dysfunction. SARS-CoV-2 infection is associated with systemic inflammation, oxidative stress, and metabolic dysregulation, all of which may impair liver function and promote glucose intolerance. This study investigated the role of SARS-CoV-2, specifically its Main Protease (M^pro^), in accelerating insulin resistance and metabolic dysfunction in HepG2 cells in vitro. **Methods:** HepG2 cells were treated with varying concentrations of M^pro^ (2.5, 5, 10, 20, 40, 80, and 160 nmol/mL) for 24 h to assess cytotoxicity and glucose uptake. Based on initial findings, subsequent assays focused on higher concentrations (40, 80, and 160 nmol/mL). The effects of M^pro^ on cell viability, protein kinase B (AKT) expression, matrix metallopeptidase-1 (MMP1), dipeptidyl peptidase 4 (DPP4), interleukin-6 (IL-6) expression, and lipid peroxidation were investigated. **Results:** Our findings reveal that the SARS-CoV-2 M^pro^ treatment led to a concentration-dependent reduction in glucose uptake in HepG2 cells. Additionally, the M^pro^ treatment was associated with reduced insulin-stimulated AKT activation, particularly at higher concentrations. Inflammatory markers such as IL-6 were elevated in the extracellular medium, while DPP4 expression was decreased. However, extracellular soluble DPP4 (sDPP4) levels did not show a significant change. Despite these changes, cell viability remained relatively unaffected, suggesting that the HepG2 cells were able to maintain overall metabolic functions under M^pro^ exposure. **Conclusions:** This study demonstrated the concentration-dependent impairment of hepatic glucose metabolism, insulin signaling, and inflammatory pathways in HepG2 cells acutely exposed to the SARS-CoV-2 M^pro^. These findings warrant further investigation to explore the long-term metabolic effects of SARS-CoV-2 and its proteases in the liver and to develop potential therapeutic approaches for post-viral metabolic complications.

## 1. Introduction

The emergence of SARS-CoV-2 has led to a global health crisis, with COVID-19 affecting multiple organ systems beyond the respiratory tract [1]. Among the less understood yet increasingly recognized complications of COVID-19 is its impact on metabolic health, particularly in the development of insulin resistance and new-onset type 2 diabetes mellitus (T2DM) [2]. With the prevalence of DM increasing worldwide, it is worthwhile to explore the pathophysiology involved in COVID-19-induced DM. Epidemiological data suggest that a growing number of COVID-19 survivors present with impaired glucose metabolism, raising concerns about the long-term consequences of SARS-CoV-2 infection on metabolic disorders [3]. While pre-existing DM is a known risk factor for severe COVID-19 outcomes, there is increasing evidence that SARS-CoV-2 itself may contribute to the onset of metabolic dysfunction, independent of prior diabetic status [2,4]. DM encompasses a group of long-term endocrine metabolic disorders marked by impaired glycaemic regulation. This results in elevated blood glucose concentrations, primarily due to insufficient insulin production, reduced insulin sensitivity, or a combination of both [5]. DM has been implicated as one of the primary causes of both morbidity and mortality in developing countries such as South Africa [6]. Globally, it is projected that the prevalence of DM will increase from 415 million individuals in 2015 to 642 million individuals by the year 2040, indicating a significant rise in the burden of DM [7].

The liver plays a central role in glucose homeostasis, regulating insulin sensitivity through key metabolic pathways such as gluconeogenesis, glycogenolysis, and lipid metabolism [8]. Insulin is a peptide hormone released by the beta cells within the Islets of Langerhans in the pancreas. It has central regulatory roles in hepatic cells [9,10]. Insulin can regulate blood glucose concentration in part through regulating gluconeogenesis and glycogenolysis in the liver [11,12]. SARS-CoV-2 has been shown to trigger a systemic inflammatory response, potentially affecting liver function and impairing insulin signaling pathways [5]. However, the precise mechanisms by which SARS-CoV-2 affects the liver remain poorly understood.

SARS-CoV-2 encodes M^pro^, which is essential for viral replication [13]. Recent findings suggest that M^pro^ may interfere with host cellular pathways [14]. Wenzel et al. reported that M^pro^ was stated to cause microvascular brain pathology through the cleaving of the nuclear factor (NF)-κB essential modulator (NEMO), the critical modulator of nuclear factor-B. The result of this elimination of NEMO is the death of human brain endothelial cells [14]. The authors also observed that M^pro^ triggered apoptosis and oxidative stress in rat cortex neurons, leading to neuronal damage [14]. Various hypotheses have been proposed based on existing data; however, the precise mechanisms behind this phenomenon remain uncertain. With this background, examining the impact of M^pro^ on key metabolic markers, oxidative stress, and hepatic insulin signaling pathways, this research seeks to identify potential mechanisms linking SARS-CoV-2 infection to liver-mediated glucose dysregulation.

## 2. Materials and Methods

### 2.1. Protein Preparations

In total, 0.2 mg of COVID-19 M^pro^ (CAE0172-200UG) was sourced from Sigma-Aldrich (St. Louis, MO, USA) in powdered form. A stock solution (5.900 nmol/mL) was prepared by dissolving M^pro^ in 1 mL of DMSO. The working concentrations of 2.5–160 nmol/mL were prepared from the stock solution via serial dilutions. The solution was filtered through a 0.22 µm membrane and stored at −20 °C. All preparations were conducted under strict aseptic conditions, utilizing a biosafety cabinet, sterile consumables, and appropriate personal protective equipment to minimize the risk of contamination.

NovoRapid^®^ insulin (100 units/mL) was used for the study. To ensure uniform distribution, the insulin vial was gently agitated before use. A stock solution of 0.05 units/mL was prepared by diluting 5 µL of insulin in 10 mL of DMEM under sterile conditions, followed by thorough mixing with a vortex mixer. The prepared solution was filtered using a 0.22 µm membrane to maintain sterility and subsequently stored at 2–8 °C until use.

### 2.2. Cell Culture Protocol

The assays were performed using HEPG2 cells sourced from Cellonex, Johannesburg, South Africa. The HEPG2 cells were cultured in DMEM supplemented with 10% FBS and 1% penicillin-streptomycin. Cells were maintained at 37 °C in a humidified incubator with 5% CO_2_. Once they reached approximately 80% confluence, they were trypsinized and transferred to new tissue culture flasks before being seeded into 24- or 96-well plates.

### 2.3. Cell Viability Assay: MTT Assay

HepG2 cells were seeded into 96-well plates at a density of 4.15 × 10^4^ cells/mL and cultured until they reached approximately 80% confluence. The cells were then treated with varying concentrations of M^pro^ (2.5, 5, 10, 20, 40, 80, and 160 nmol/mL), while control wells contained only the culture medium without M^pro^. Following treatment, the cells were incubated at 37 °C for 24 h. After incubation, the culture medium was aspirated, and the MTT solution was prepared by dissolving 1 mg of MTT powder in a solution consisting of 10% PBS and 90% FBS-free media. The solution was filter-sterilized, and 100 μL was added to each well. The plates were then incubated in the dark at 37 °C for 3 h to allow the formation of formazan crystals. The MTT solution was subsequently discarded, and 100 μL of DMSO was added to each well to dissolve the crystals. The plates were further incubated for 10 min, after which the absorbance was measured at 570 nm using a spectrophotometer. The assay was conducted in duplicate, with each concentration tested in sextuplicate to ensure reproducibility. Cell viability was calculated as a percentage of the absorbance of the control group using the following equation:$$Percentage\;cell\;viability\;=\left(\frac{Average\;absorbance\;of\;treated\;wells}{Average\;absorbance\;of\;control}\right)\;\times 100$$

### 2.4. Estimated Glucose Utilization Assay

The glucose utilization assay was performed with modifications to an established protocol by Ventera et al. with some modifications [15]. HepG2 cells (3.79 × 10^4^ cells/mL) were seeded into 24-well plates and incubated in EMEM at 37 °C with 5% CO_2_ until they reached approximately 80% confluence. Following cell preparation, the culture media were removed, and 250 µL of M^pro^ at concentrations of 2.5, 5, 10, 20, 40, 80, and 160 nmol/mL was added to the designated wells. A control group received only fresh culture media. Baseline glucose concentrations (t = 0 h) were recorded, and the cells were incubated for 24 h, after which glucose levels in the media were measured again. To assess the effect of M^pro^ on insulin sensitivity, cells were first exposed to M^pro^ for 24 h, followed by treatment with 250 µL of insulin-containing media (0.05 units/mL). Media glucose levels were measured at t = 0 and 24 h using a glucometer. Each concentration was tested in quadruplicate, and the experiment was repeated three times for consistency and reliability. The estimation of glucose uptake was determined using the following equation:$$Glucose\;uptake\;(\%)=\left(\frac{Medium\;glucose\left(T0\right)-Medium\;glucose(T24)}{Medium\;glucose(T0)}\right)\times 100$$

### 2.5. In Cell ELISA: Semi-Quantitative Expression of AKT, DPP4, MMP1, and IL-6

HepG2 cells were seeded into 96-well plates and cultured in DMEM for ELISA experiments. After 24 h, the cells were treated with 100 µL of M^pro^ at concentrations of 40, 80, and 160 nmol/mL and incubated overnight. On the following day, the media was aspirated and stored at 4 °C in vials. Control wells included a blank control, consisting of DMEM without cells or M^pro^, and a negative control, which contained cells treated with primary and secondary antibodies but without M^pro^. The blank control assessed the background signal from DMEM and potential reagent contaminants, while the negative control accounted for any non-specific binding of antibodies in the absence of M^pro^. Then, 100 µL of 8% paraformaldehyde was added to each well to fix and cross-link the cells to the plate. The plate was shaken at 300 rpm and at room temperature for 15 min. The fixative was then aspirated, followed by three washes with 200 µL of 1X PBS per well. After the final wash, the plate was blotted dry. Next, cells were permeabilized using 250 µL of permeabilization buffer (Triton X-100 in 1X PBS) for 30 min at room temperature while shaking. The permeabilization buffer was then aspirated, and 200 µL of blocking buffer (prepared by dissolving 2 g of bovine serum albumin in 20 mL of 1X PBS) was added. The plate was incubated for 2 h at room temperature with shaking at 300 rpm.

Following blocking, 100 µL of primary antibodies against protein kinase B (AKT), matrix metalloproteinase-1 (MMP1), dipeptidyl peptidase-4 (DPP4), and interleukin-6 (IL-6), each diluted 1:5000, was added to designated wells for each in the cell-based ELISA experiment. The plate was incubated overnight at 4 °C. On the next day, the primary antibody was aspirated, and wells were washed three times with 250 µL of the wash buffer (prepared by adding 625 µL Tween-20 to 250 mL of 1X PBS) to remove unbound antibodies. A 100 µL solution of secondary antibody (anti-rabbit IgG, diluted 1:5000) was then added to each well, followed by incubation at room temperature for 2 h with gentle shaking. After incubation, wells were washed four times with 250 µL of wash buffer before adding 100 µL of TMB horseradish peroxidase substrate. The plate was incubated in the dark for 30 min. To stop the reaction, 100 µL of 0.1M HCl was added, inducing a color change from blue to yellow. Absorbance was measured at 450 nm using a UV-VIS spectrophotometer. The assay was conducted in duplicates, with each concentration tested in triplicate to ensure reproducibility. The relative expression of each protein was calculated as follows:$$Relative\;percentage\;expression=\left(\frac{Average\;absorbance\;of\;treated\;wells}{Average\;absorbance\;of\;control}\right)\times 100$$

### 2.6. Extracellular Protein Analysis: IL-6, MMP1, and DPP4

To assess the extracellular levels of IL-6, DPP4, and MMP1, the collected media were centrifuged at 300–500×*g*, and the supernatant (50 µL) was carefully transferred into a high-binding plate for analysis and incubated for 6 h at room temperature. After incubation, the media was removed, and the plate was gently blotted dry to eliminate any residual liquid. Unlike the in-cell ELISA, fixation and permeabilization steps were not required. However, all subsequent ELISA procedures, including antibody incubation, washing, and detection, were performed as previously described. The relative protein expression for each target was calculated using the formula above.

### 2.7. Lipid Peroxidation: Media Malondialdehyde Concentration

Following cell treatment with Mpro, 250 µL of the supernatant was aspirated and used for the TBARS assay. The harvested media were mixed with 400 µL of 2% phosphoric acid, and the resultant solution was further aliquoted into two separate tubes. Afterwards, phosphoric acid (200 μL of 7%) was added into both glass tubes, followed by the addition of HCL (400 µL, 3 mM) into one glass test tube (blank) and 400 μL of thiobarbituric acid (TBA)/butylated hydroxytoluene (BHT) into the other test tube (sample test). To achieve an acidic pH of 1.5, 200 μL of 1M HCl was added to both the blank and sample test tubes. All test tubes were boiled at 100 °C for 15 min using the Labotec shaking Eco Bath water bath. They were then left to cool to room temperature for 10 min. After cooling, 1.5 mL of butanol was added to each test tube, which was then vortexed for 10 s. The test tubes were left to stand until the butanol phase (the top layer) became visible. Then, 200 mL of the butanol phase was added in triplicate to a 96-well plate. The optical density of the samples was measured spectrophotometrically at 532 nm, with a reference wavelength of 600 nm, using the Varioskan Lux microplate reader. MDA concentrations (mM) were calculated by dividing the average absorbance of the samples by the absorption coefficient (156 Mm^−1^). The absorbance from these wavelengths was used to calculate the concentration of MDA using the Beer–Lambert law as follows:$$MDA\;concentration=\left(\frac{Final\;absorbance\;-\;Blank}{MDA\;coefficient\left(156\;{mmol}^{-1}\right)}\right)$$

While MDA concentrations were calculated based on the Beer–Lambert law, the results are presented as percentage changes relative to the control group to highlight oxidative stress induction across treatments

### 2.8. Statistical Analysis

Data are presented as mean ± standard deviation (SD) from three independent replicates. Graphs were created using GraphPad Prism (Version 8, GraphPad Software, San Diego, CA, USA, 2019). Normality was evaluated through normality and log-normality tests in GraphPad Prism. For normally distributed data, one-way ANOVA followed by Tukey’s multiple comparison test was used to compare the effects of M^pro^ concentrations with the control group. For non-normally distributed data, the Kruskal–Wallis test was applied. Statistical significance was set at *p* ≤ 0.05, with asterisks (*) indicating significant differences between the control and treatment groups. Despite the use of a non-parametric test for non-normally distributed data, the mean and standard deviation were reported for consistency across all datasets.

## 3. Results

### 3.1. Cytotoxic Assay

A cytotoxicity assessment was performed on the HepG2 cell line using M^pro^ at concentrations of 2.5, 5, 10, 20, 40, 80, and 160 nmol/mL, as illustrated in Figure 1. Overall, the findings demonstrated that M^pro^ did not exhibit cytotoxic effects on hepatic cells within this concentration range. Although the lower concentrations (2.5–20 nmol/mL) resulted in a reduction in cell viability, the decrease was not statistically significant compared to the untreated control. In contrast, treatment with 40 nmol/mL and 80 nmol/mL led to a significant decline in cell viability relative to the control.

### 3.2. Effect of M^pro^ on Non-Insulin Stimulated Glucose Uptake

The relative glucose uptake was evaluated in HepG2 cells following a 24 h treatment with varying concentrations of M^pro^ (2.5, 5, 10, 20, 40, 80, and 160 nmol/mL). The findings, depicted in Figure 2, revealed that the insulin-treated control (0′) exhibited a greater glucose uptake compared to the untreated control (0), confirming insulin’s expected role in enhancing glucose transport. This response aligns with insulin’s well-established function in facilitating cellular glucose uptake, thereby validating both the experimental conditions and the sensitivity of the HepG2 cells to insulin stimulation. In HepG2 cells, the treatment with low M^pro^ concentrations (2.5 to 40 nmol) had no effect on glucose uptake by comparison to the non-treated control. However, the 80 nmol/mL and 160 nmol/mL concentrations of M^pro^ showed a significant decline by comparison to the non-treated control. The glucose uptake results show a concentration-dependent reduction in glucose uptake across the different concentrations tested (2.5 to 160 nmol/mL). As the concentration of M^pro^ increases from 2.5 to 160 nmol/mL, glucose uptake progressively declines, with the most substantial reduction observed at the highest concentrations (80 and 160 nmol/mL).

### 3.3. Effect of M^pro^ on Insulin-Stimulated Glucose Uptake

Figure 3 below represents the effect of SARS-CoV-2 M^pro^ on insulin-stimulated glucose uptake in HepG2 cells. The control group (0) shows baseline glucose uptake, while the insulin control group (0′) demonstrated an increase in glucose uptake, confirming insulin responsiveness in these cells. Pre-treatment of HEPG2 cells with lower concentrations of M^pro^ (2.5–20 nmol/mL) did not significantly alter insulin-stimulated glucose uptake. However, pre-treatment of HEPG-2 cells with higher concentrations (40, 80, and 160 nmol/mL) demonstrated a significant reduction in glucose uptake. This suggests that M^pro^ may impair insulin signaling and glucose metabolism in HepG2 cells in a concentration-dependent manner, potentially contributing to insulin resistance mechanisms.

### 3.4. Effect on AKT Expression

Figure 4 illustrates the relative AKT expression in HepG2 cells following treatment with increasing concentrations of SARS-CoV-2 M^pro^ (40, 80, and 160 nmol/mL) compared to the control (untreated cells). AKT expression remains relatively unchanged across all treatment groups, with no significant reduction observed. This suggests that, while M^pro^ impairs insulin-stimulated glucose uptake at higher concentrations, it does not significantly alter total AKT expression in HepG2 cells.

### 3.5. Effect on Insulin-Stimulated AKT Expression

Figure 5 illustrates the effect of SARS-CoV-2 M^pro^ on insulin-stimulated AKT expression in HepG2 cells. Insulin treatment (0′) slightly increased AKT expression compared to the control (0), confirming its role in insulin signaling activation. However, M^pro^ pre-treatment at 40, 80, and 160 nmol/mL leads to a progressive, significant reduction in AKT expression, with higher concentrations showing greater inhibition (** *p* < 0.01 at 40 nmol/mL; **** *p* < 0.0001 at 80 and 160 nmol/mL).

### 3.6. Relative Cellular and Media DPP4

The relative DPP4 expression in HepG2 cells following treatment with increasing concentrations of SARS-CoV-2 M^pro^ (40, 80, and 160 nmol/mL) compared to the control is shown in Figure 6 and Figure 7. A dose-dependent increase in DPP4 expression is observed, with the highest concentration (160 nmol/mL) showing significant upregulation (**** *p* < 0.0001). Figure 7 represents the relative DPP4 levels in the culture medium of HepG2 cells following treatment with increasing concentrations of SARS-CoV-2 M^pro^ (40, 80, and 160 nmol/mL) compared to the control. Unlike intracellular DPP4 expression, there is no significant change in the levels of shed DPP4 in the medium across different M^pro^ concentrations. This suggests that while M^pro^ upregulates cellular DPP4 expression, it does not significantly enhance DPP4 shedding into the extracellular environment.

### 3.7. Cellular and Medium IL-6

Figure 8 illustrates the relative in-cell expression percentage of IL-6 in the HepG2 cell lines. The HepG2 cells treated with M^pro^ exhibit a modest increase in IL-6 expression, with the 80 and 160 nmol/mL concentrations showing no statistical difference compared to the control. Figure 9 presents the relative IL-6 levels in the culture medium of HepG2 cells following treatment with increasing concentrations of SARS-CoV-2 M^pro^ (40, 80, and 160 nmol/mL) compared to the control. A concentration-dependent increase in IL-6 secretion was observed, with significantly elevated levels at 80 and 160 nmol/mL (*** *p* < 0.001). This suggests that M^pro^ induces a pro-inflammatory response in HepG2 cells, leading to increased IL-6 release.

### 3.8. Relative Cellular and Media MMP1

Figure 10 depicts the relative cellular expression of MMP1 in HepG2 cells was assessed after treatment with M^pro^ at 40, 80, and 160 nmol/mL. The results show no significant change in MMP1 expression from the control in the HepG2 cell line for all concentrations of M^pro^ when compared to the control (untreated cells). No statistically significant difference was observed for the HepG2 cell line. Figure 11 depicts the relative MMP1 levels in the culture medium of HepG2 cells following treatment with increasing concentrations of SARS-CoV-2 M^pro^ (40, 80, and 160 nmol/mL) compared to the control. A concentration-dependent increase in MMP1 secretion is observed, with a significant rise at 160 nmol/mL (* *p* < 0.05).

### 3.9. Relative MDA Concentration

To evaluate the oxidative stress status, malondialdehyde (MDA) levels were quantified in cells exposed to increasing concentrations (40, 80, and 160 nmol/mL). The results, as shown in Figure 12 indicate a concentration-dependent increase in MDA concentrations compared to the untreated control. At 40 nmol/mL and 80 nmol/mL, the MDA levels exhibited a slight increase relative to the control, but the differences were not statistically significant. However, at 160 nmol/mL, a significant elevation in MDA levels was observed (*p* < 0.05), as indicated by the asterisk in Figure 12. This suggests that higher concentrations of M^pro^ induce oxidative stress, leading to increased lipid peroxidation.

## 4. Discussion

The liver has a regenerative ability and carries out a wide range of complex functions, such as replenishing and storing easily accessible energy in the form of glycogen, regulating carbohydrate metabolism, and detoxifying and eliminating metabolic byproducts from the body [16]. Despite their cancerous origin, HepG2 cells retain several key characteristics of differentiated hepatocytes, including albumin secretion, insulin-stimulated glycogen synthesis, and glutathione-based detoxification [17,18]. Due to its broad applications in scientific research, such as in pharmacological and toxicological studies, the HepG2 cell line is highly popular [18]. Additionally, they have a stable phenotype, normal liver cell genetic characteristics, high glucose consumption, and active energy metabolism [19].

The liver is vital for metabolic balance, detoxification, and immune function, but these essential roles also make it vulnerable to stressors that can cause cell damage and dysfunction [20]. Findings from the post-mortem analyses of individuals infected with SARS-CoV-2 have identified traces of viral proteins, suggesting that the virus may exhibit hepatotropic characteristics [21]. In the cytotoxic study, the viability of the cells decreased slightly; however, the reduction was not significant as cells retained viable metabolism. Our findings were consistent with the study conducted by Barreto et al., who reported that infection with SARS-CoV-2 in hepatocytes releases infectious viral particles without damaging cells [22].

The liver also plays an important role in glucose metabolism. In the liver, insulin activates insulin receptor tyrosine kinase (IRTK), leading to the phosphorylation of insulin receptor substrate-1 (IRS1) and insulin receptor substrate-2 (IRS2) and the activation of AKT2 [23]. This subsequently reduces hepatic glucose production, enhances glycogen synthesis, and stimulates lipogenesis [12]. Furthermore, insulin signaling, especially through the PI3K/AKT pathway, promotes GLUT2 activity and translocation in hepatocytes, enabling glucose uptake [23]. A disruption in hepatic glucose production has been shown to exacerbate T2DM hyperglycaemia [8]. In our study, M^pro^ treatment caused a concentration-dependent reduction in baseline glucose uptake across the different concentrations of M^pro^ utilized (2.5 to 160 nmol/mL). Although GLUT2 is the principal glucose transporter in hepatocytes, accounting for more than 97% of glucose flux across the plasma membrane, unlike GLUT4 in skeletal muscle and adipose tissue, GLUT2 is not directly regulated by insulin [23]. Instead, in hepatocytes, insulin exerts its primary effects via intracellular signaling cascades that regulate key metabolic pathways such as glycogen synthesis and gluconeogenesis [23,24]. These include the IRS/PI3K/AKT2/GSK3 pathway that regulates glycogenesis, gluconeogenesis, and lipogenesis and the IRS/PI3K/AKT2/FOXO1 axis, which suppresses gluconeogenic gene expression. Importantly, both pathways are sensitive to inhibition by cellular stress signals and proinflammatory mediators [25]. The observed reduction in glucose uptake may be attributed to M^pro^-induced oxidative stress or inflammatory signaling, which can impair GLUT2 function or reduce its membrane localization, rather than direct effects on GLUT2-mediated transport.

In line with these findings, M^pro^ exposure was associated with altered insulin signaling in HepG2 cells, with more pronounced changes observed at higher concentrations (40–160 nmol/mL). An investigation into the exact position of the disruption led us to assess baseline and insulin-stimulated AKT expression. Our findings showed no significant change in baseline AKT expression. However, insulin-stimulated AKT phosphorylation was significantly reduced at higher M^pro^ concentrations (40–160 nmol/mL), suggesting that M^pro^ may affect insulin signaling under stimulated conditions. This aligns with previous studies showing that viral proteins can disrupt insulin pathways through oxidative stress, inflammation, and disturbances in insulin receptor substrate activity [26,27,28,29]. While these alterations provide insights into M^pro’s^ metabolic impact, further investigation is needed to delineate whether these effects are transient stress responses or sustained impairments in insulin signaling.

Building on the observed inhibitory effect of M^pro^ on insulin-stimulated AKT activation, we further examined oxidative stress status in HepG2 cells by measuring malondialdehyde (MDA) levels, a key marker of lipid peroxidation. Our results indicate a dose-dependent increase in MDA levels with rising M^pro^ concentrations, suggesting that oxidative stress is a contributing factor to the observed disruptions in insulin signaling. Oxidative stress is known to play a major role in metabolic dysfunction, with increased lipid peroxidation products like MDA being linked to problems with insulin signaling and β-cell function [30]. These findings are in line with previous research showing that SARS-CoV-2 infection can cause oxidative stress in different cell types, reinforcing the idea that viral infections contribute to metabolic disturbances [31,32]. Going forward, it would be valuable to investigate whether antioxidants or ROS inhibitors can help reduce M^pro^-induced oxidative stress and protect insulin signaling in hepatocytes.

COVID-19 has also been associated with inflammatory processes involving key mediators such as DPP4 and MMP1, which are known to mediate both tissue remodeling and systemic inflammation [33,34]. Viruses can trigger chronic inflammation, which may lead to cellular dysfunction and even transformation [35]. A study conducted by Lepiller and colleagues demonstrated that the infection of HepG2 cells with human cytomegalovirus (HCMV) caused the production of IL-6 in the supernatants of HepG2 cells, starting as early as 2 h post-infection [36]. Similarly, Carlquist et al. reported increased IL-6 gene transcription and protein expression in human lung fibroblasts following exposure to HCMV [37]. This suggests that during viral infections, the liver may act as a cytokine producer, contributing to systemic inflammation. Our data shows that M^pro^ induced an increase in extracellular IL-6 levels in the HepG2 cell line, corroborating previous findings that viral proteins can stimulate hepatic cytokine release [36,37]. Inflammation is a well-established contributor to insulin resistance, offering a potential mechanism through which M^pro^ may contribute to glycaemic dysregulation. In addition to IL-6, we observed a significant increase in MMP1 levels. While MMP1 is classically known for degrading extracellular matrix components, it also exerts pro-inflammatory effects via the activation of protease-activated receptor-1 (PAR1) on hepatocytes [38]. Activation of PAR1 has been implicated in hepatic fibrosis, cytokine release, and further amplification of inflammatory signaling [38]. Therefore, the upregulation of MMP1 following M^pro^ exposure may contribute to liver inflammation and insulin resistance not only through matrix remodelling but also via PAR1-mediated pro-inflammatory pathways. Given that enzymes like DPP4 and MMP1 play critical roles in immune regulation and metabolic processes, their dysregulation in the context of viral infection could be central to the metabolic disturbances observed in COVID-19 patients.

DPP4 is a serine protease that exists in two forms: a soluble form, sDPP4, predominantly found in plasma and interstitial fluid and a membrane-bound form expressed in vascular smooth muscle, endothelial cells, blood, kidneys, and intestines [33,39,40]. While much of the research on DPP4 has focused on its role in T2DM treatment, studies have also suggested that DPP4 functions as an adipokine [22,40,41,42]. Although few studies have directly explored the impact of viral proteins or proteases on DPP4 shedding, MMPs, including MMP1, are known to regulate DPP4 cleavage and activity [33].

Liver damage and its subsequent regeneration are intricately connected to complex extracellular matrix signaling pathways, which are influenced by proteases like MMP1 [43]. Our data showed high levels of extracellular MMP1, which would typically correlate with increased sDPP4. However, we observed a decrease in sDPP4, suggesting that M^pro^ exposure may alter DPP4 shedding dynamics, potentially by affecting the cleavage process or shifting cellular priorities under stress. In hepatocytes, pro-inflammatory or fibrogenic stimuli like IL-6 or transforming growth factor (TGF-β) can upregulate MMP1 expression [34]. Although extracellular IL-6 levels were elevated in our study, intracellular IL-6 expression remained low in HepG2 cells, which may explain why MMP1 levels were not as high as expected, potentially limiting DPP4 shedding.

Interestingly, M^pro^-induced stress may prompt a prioritisation of acute-phase protein synthesis over the cleavage and release of proteins such as DPP4. During metabolic stress or inflammation, hepatocytes often shift towards producing inflammatory mediators to manage systemic responses [44]. In the liver, decreased sDPP4 might serve as a regulatory mechanism to limit excessive inflammation. Since sDPP4 is associated with immune activation, HepG2 cells may retain DPP4 to avoid exacerbating inflammation, especially if extracellular IL-6 is already elevated [33]. Therefore, the increase in intracellular DPP4 in HepG2 cells could be a protective mechanism. By limiting DPP4 shedding, HepG2 cells may be attempting to control sustained systemic inflammation.

While this study provides important insights into the potential effects of SARS-CoV-2 Mpro on hepatic insulin signalling, oxidative stress, and inflammatory responses, several limitations should be considered when interpreting the findings. Firstly, in a physiological SARS-CoV-2 infection, M^pro^ functions intracellularly. Therefore, the observed effects may not result from direct interference with insulin signaling by M^pro^ but rather from stress responses triggered by protease exposure. These include oxidative stress and the increased production of proinflammatory cytokines such as IL-6 and MMP1. It is well known that stress-activated pathways can impair insulin receptor function and downstream signaling. Secondly, all experiments were conducted using HepG2 cells, a hepatocellular carcinoma-derived cell line, which may not fully replicate the physiological responses of healthy hepatocytes. Future studies using more physiologically relevant models, such as intracellular M^pro^ expression systems or viral infection models, are needed to further elucidate the mechanisms involved.

Ultimately, these findings suggest that SARS-CoV-2 M^pro^ may contribute to hepatic insulin resistance, which could help explain the metabolic disturbances observed in some COVID-19 patients. The disruption of insulin signaling, combined with proinflammatory and oxidative effects, supports the hypothesis that M^pro^ could play a role in the development of post-infection glucose dysregulation. Further studies using animal models and clinical samples are needed to determine whether these cellular changes contribute to the increased incidence of new-onset diabetes following COVID-19. Understanding the metabolic impact of viral proteases like M^pro^ may offer new insights into the prevention and management of long-term complications in COVID-19 survivors.

## 5. Conclusions

This study demonstrates that SARS-CoV-2 M^pro^ exerts multiple effects on HepG2 cells, influencing glucose uptake, insulin signaling, and inflammatory responses. The observed inhibition of baseline glucose uptake, disruption of insulin-stimulated AKT activation, and alterations in IL-6 and DPP4 expression suggest that M^pro^ may contribute to hepatic metabolic dysregulation. While our data provide insights into the potential mechanisms by which M^pro^ affects liver cells, further studies are necessary to fully elucidate its role in hepatic glucose metabolism and its implications for post-COVID-19 metabolic complications.

## Figures and Tables

**Figure 1 pathophysiology-32-00039-f001:**
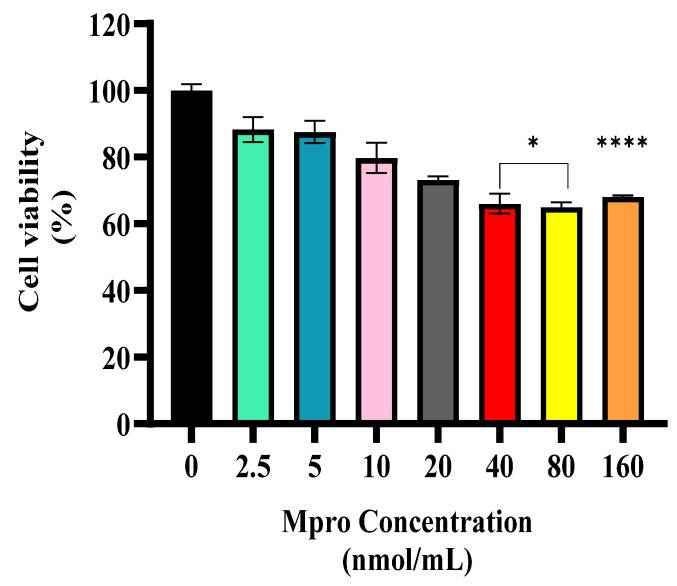
Cell viability was assessed in HepG2 cells following a 24 h treatment with varying concentrations of M^pro^ (2.5, 5, 10, 20, 40, 80, and 160 nmol/mL). The results are presented as a percentage, with untreated cells serving as the control group. Data are expressed as mean ± standard deviation, with error bars indicating variability (*n* = 3). Statistical significance between the treated groups and the control is denoted by asterisks: (*) *p* < 0.05 and (****) *p* < 0.0001.

**Figure 2 pathophysiology-32-00039-f002:**
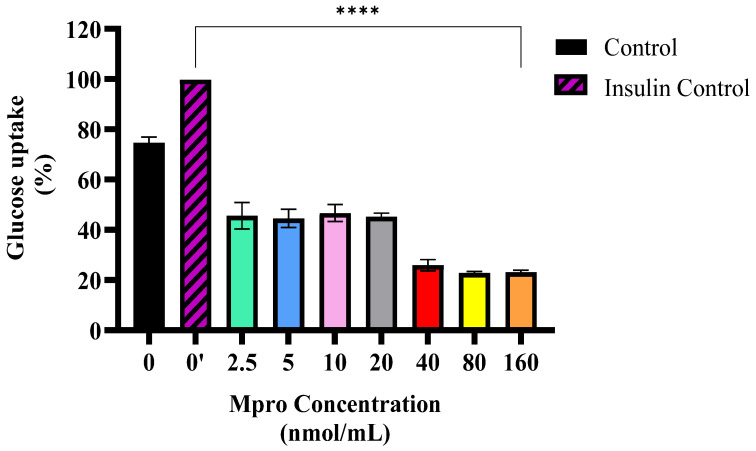
The relative glucose uptake was assessed in HepG2 cells following a 24 h treatment with varying concentrations of M^pro^ (2.5, 5, 10, 20, 40, 80, and 160 nmol/mL). The results are presented as a percentage, with untreated cells serving as the control group. Untreated cells are presented as 0, while 0′ represents insulin-treated cells. Untreated cells were used as the control group. Data are expressed as mean ± standard deviation, with error bars indicating variability (*n* = 3). Statistical significance between the treated groups and the control is denoted by asterisks: (****) *p* < 0.0001.

**Figure 3 pathophysiology-32-00039-f003:**
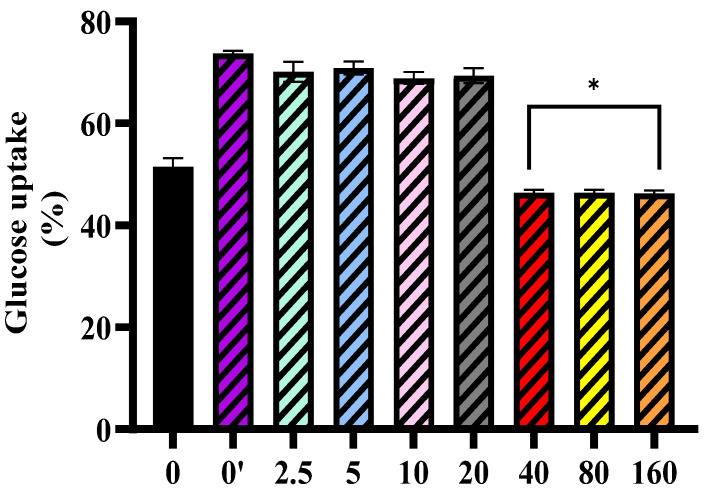
Insulin-stimulated glucose uptake was assessed in HepG2 cells following a 24 h treatment with varying concentrations of M^pro^ (2.5, 5, 10, 20, 40, 80, and 160 nmol/mL). Untreated cells are presented as 0, while 0′ represents insulin-treated cells. The results are presented as a percentage, with untreated cells serving as the control group. Data are expressed as mean ± standard deviation, with error bars indicating variability (*n* = 3). Statistical significance between the treated groups and the control is denoted by asterisks: (*) *p* < 0.05.

**Figure 4 pathophysiology-32-00039-f004:**
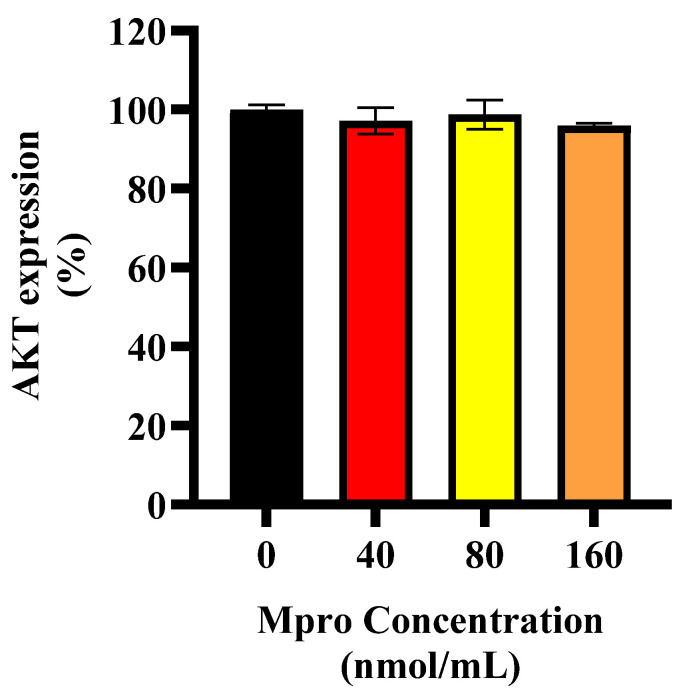
AKT expression was assessed in HepG2 cells following a 24 h treatment with varying concentrations of M^pro^ (40, 80, and 160 nmol/mL). The results are presented as a percentage, with untreated cells serving as the control group. Data are expressed as mean ± standard deviation, with error bars indicating variability (*n* = 3). No statistically significant difference was observed.

**Figure 5 pathophysiology-32-00039-f005:**
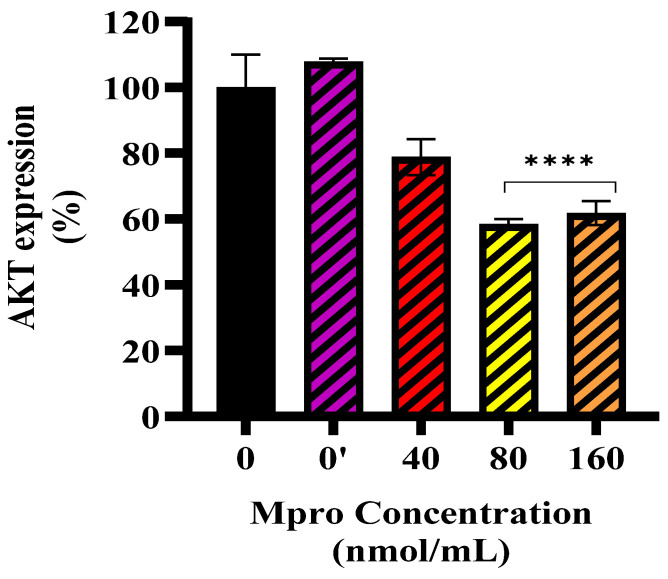
Insulin-stimulated AKT expression was assessed in HepG2 cells following a 24 h treatment with varying concentrations of M^pro^ (40, 80, and 160 nmol/mL). Results are expressed as a percentage. Untreated cells are presented as 0, while 0′ represents insulin-treated cells. Insulin-treated cells were used as the control group. Data are expressed as mean ± standard deviation, with error bars indicating variability (*n* = 3). Statistical significance between the treated groups and the control is denoted by asterisks: (****) *p* < 0.0001.

**Figure 6 pathophysiology-32-00039-f006:**
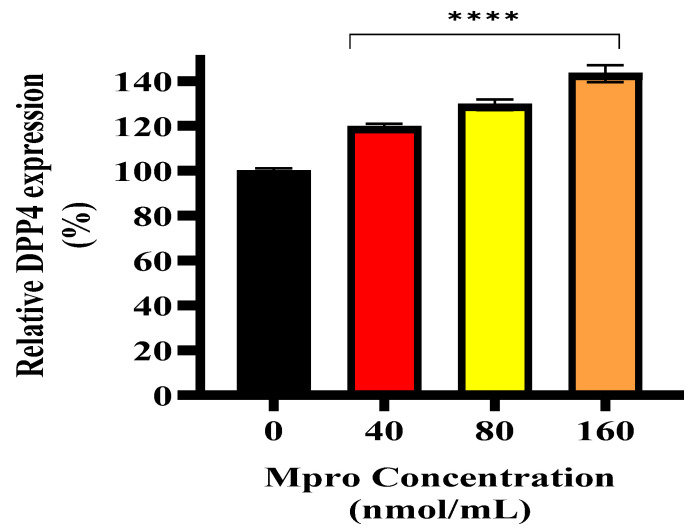
Relative DPP4 expression was assessed in HepG2 cells following a 24 h treatment with varying concentrations of M^pro^ (40, 80, and 160 nmol/mL). The results are presented as a percentage, with untreated cells serving as the control group. Data are expressed as mean ± standard deviation, with error bars indicating variability (*n* = 3). Statistical significance between the treated groups and the control is denoted by asterisks (****) *p* < 0.0001.

**Figure 7 pathophysiology-32-00039-f007:**
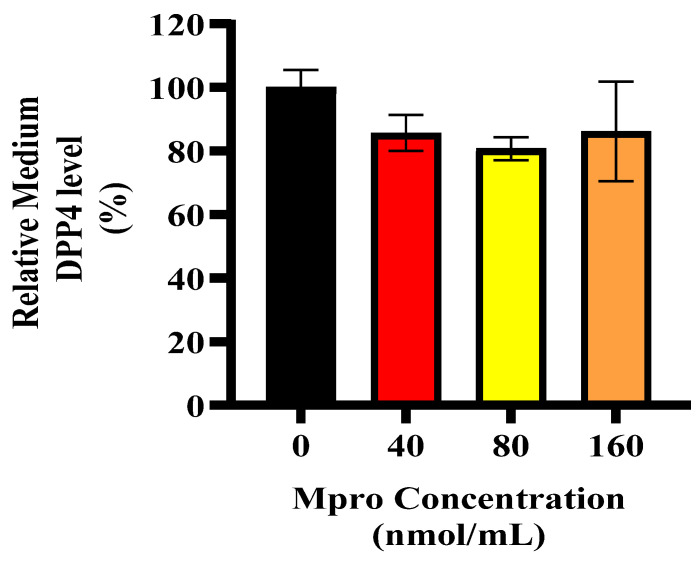
Relative DPP4 medium level was assessed in HepG2 cells following a 24 h treatment with varying concentrations of M^pro^ (40, 80, and 160 nmol/mL). The results are presented as a percentage, with the untreated cellular medium serving as the control group. Data are expressed as mean ± standard deviation, with error bars indicating variability (*n* = 3). No statistical significance between the treated groups and the control was noted.

**Figure 8 pathophysiology-32-00039-f008:**
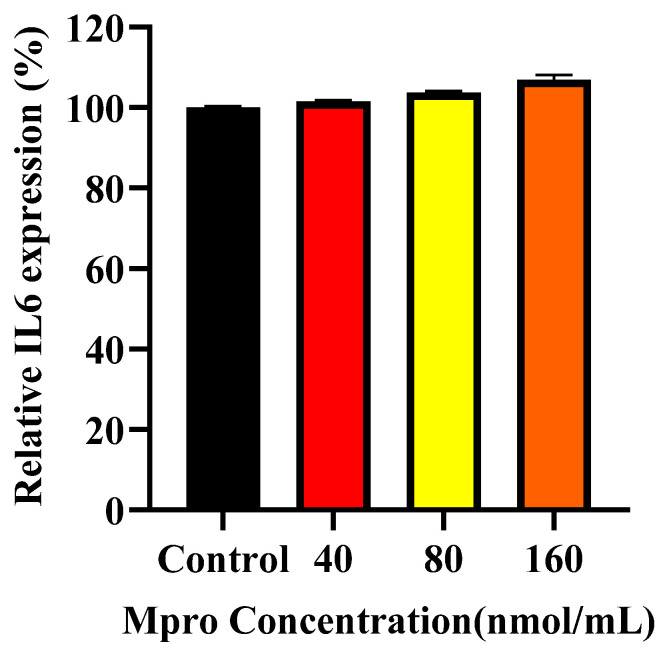
Relative IL-6 cellular expression was assessed in HepG2 cells following a 24 h treatment with varying concentrations of M^pro^ (40, 80, and 160 nmol/mL). The results are presented as a percentage, with untreated cells serving as the control group. Data are expressed as mean ± standard deviation, with error bars indicating variability (*n* = 3). No statistically significant differences between the treated groups and the control were noted.

**Figure 9 pathophysiology-32-00039-f009:**
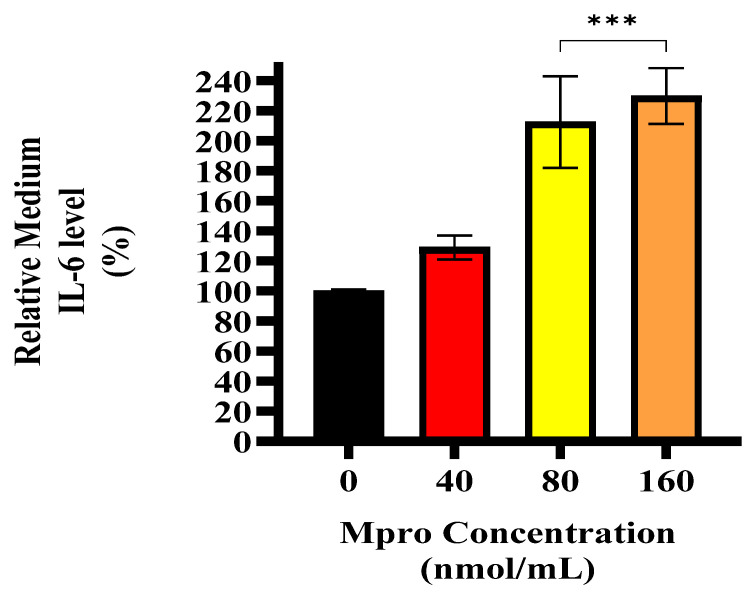
Relative IL-6 medium level was assessed in HepG2 cells following a 24 h treatment with varying concentrations of M^pro^ (40, 80, and 160 nmol/mL). The results are presented as a percentage, with untreated cells serving as the control group. Data are expressed as mean ± standard deviation, with error bars indicating variability (*n* = 3). Statistical significance between the treated groups and the control is denoted by asterisks: (***) *p* < 0.001.

**Figure 10 pathophysiology-32-00039-f010:**
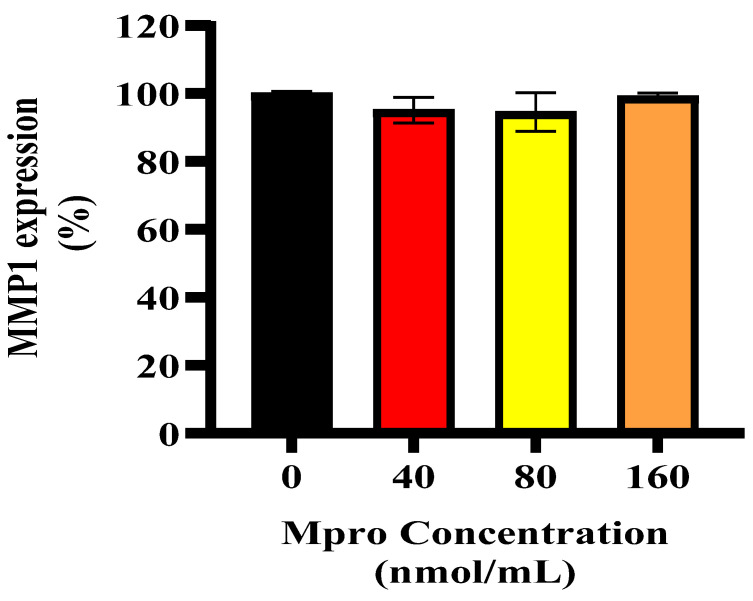
Relative MMP1 expression was assessed in HepG2 cells following a 24 h treatment with varying concentrations of M^pro^ (40, 80, and 160 nmol/mL). The results are presented as a percentage, with untreated cells serving as the control group. Data are expressed as mean ± standard deviation, with error bars indicating variability (*n* = 3). No statistically significant difference between the treated groups and the control was observed.

**Figure 11 pathophysiology-32-00039-f011:**
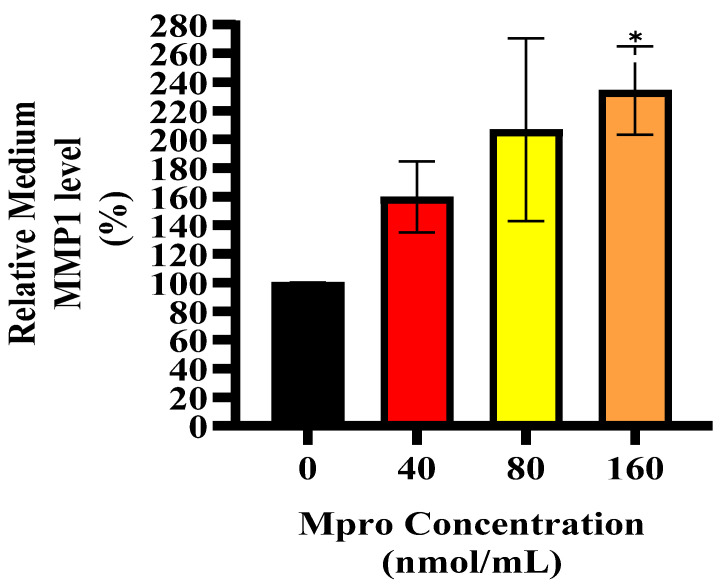
Relative MMP1 medium level was assessed in HepG2 cells following a 24 h treatment with varying concentrations of M^pro^ (40, 80, and 160 nmol/mL). The results are presented as a percentage, with untreated cells serving as the control group. Data are expressed as mean ± standard deviation, with error bars indicating variability (*n* = 3). Statistical significance between the treated groups and the control is denoted by asterisks: (*) *p* < 0.05.

**Figure 12 pathophysiology-32-00039-f012:**
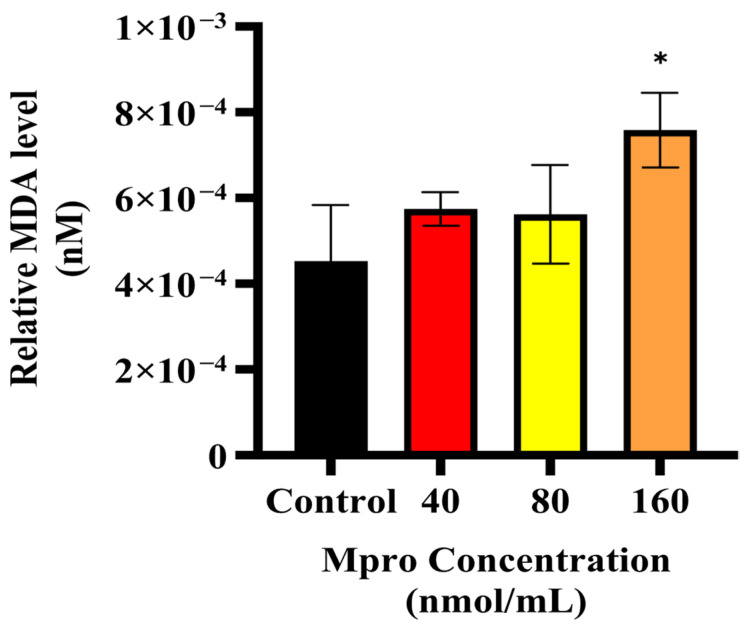
Relative MDA medium level was assessed in HepG2 cells following a 24 h treatment with varying concentrations of M^pro^ (40, 80, and 160 nmol/mL). The results are presented as a percentage, with untreated cells serving as the control group. Data are expressed as mean ± standard deviation, with error bars indicating variability (*n* = 3). Statistical significance between the treated groups and the control is denoted by asterisks: (*) *p* < 0.05.

## Data Availability

The original contributions presented in this study are included in the article. Further inquiries can be directed to the corresponding author(s).

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
