# Peer review of "SARS-CoV-2 Main Protease Dysregulates Hepatic Insulin Signaling and Glucose Uptake: Implications for Post-COVID-19 Diabetogenesis"

_pathophysiology, 2025, doi:10.3390/pathophysiology32030039_

Round 1

Reviewer 1 Report

Comments and Suggestions for Authors

In this study the authors assessed the concentration-dependent in vitro acute effects (24 h treatment of HepG2 cell line) of the SARS-CoV-2 main protease, Mpro (chymotrypsin-like protease) on: cell viability, insulin-dependent and independent glucose uptake, cellular expression of Akt, DPP4, MMP1, IL-6; the levels of DPP4, MMP1, IL-6 as well as of MDA (a marker of lipid peroxidation) were also assessed in the extracellular milieu. While no cytotoxic effects were reported at 24 h, a significant concentration-dependent reduction in glucose uptake and insulin-stimulated Akt activation for the higher concentrations (40, 80 and 160 nmol/mL) was found. In the extracellular milieu, the highest concentration (160 nmol/mL) of Mpro significantly increased the MMP1, IL-6 and MDA levels, whereas a non-significant decrease in DPP4 was noticed.

MAJOR CHANGES

Title

Please rephrase the title in order to match the paper results.

Suggestions

Characterization of the acute metabolic, proinflammatory and prooxidant effects of the SARS-CoV-2 Mpro on the HepG2 cell line

or

Metabolic impairment, proinflammatory and prooxidant effects of the SARS-CoV-2 Mpro in HepG2 cells

or alternatives...

Abstract

Please rephrase the sentence at line 28 in order to mention the concentration-dependent effect of the acute Mpro exposure:

Suggestion: This study demonstrated the concentration dependent-impairment of hepatic glucose metabolism, insulin signaling, and inflammatory pathways in HepG2 cells acutely exposed to the SARS-CoV-2 Mpro.

Results

The statistical significance should be redrawn for all the graphs in order to match – as described - the comparison between study groups and control.

The caption of Figure 3.8 should be corrected – it is the IL-6 expression (not medium level). More important, no significant differences are present among the groups, therefore I believe the mark **** was erroneously placed.

At subchapter 3.9 why is the relative MDA concentration?

Conclusion

Please shorten the conclusions, simply presenting the major findings (the last paragraph in the Discussion section is, in fact, the conclusion). Speculation regarding the potential significance for insulin resistance should be placed in the end of the Discussion.

MINOR CHANGES

Introduction

Line 40  - please change the word prudent with worthwhile.

Please define the acronyms, eg DPP4, MMP1, where they firstly appear in the body text.

Reviewer 2 Report

Comments and Suggestions for Authors

At present, there is ample evidence to suggest that insulin resistance is associated with low-grade inflammation in organs such as skeletal muscles, adipose tissue, and the liver. Conversely, the development of insulin resistance is a pivotal factor in the pathogenesis of diseases such as metabolic syndrome and type 2 diabetes mellitus. In addition to metabolic inducers of inflammation, infectious factors, including viruses, have been demonstrated to contribute to insulin resistance. In their work, the authors demonstrated the negative effect of the Main Protease (Mpro) of the SARS-CoV-2 virus on glucose uptake in vitro by human hepatocellular carcinoma (HepG2) cells. From the aforementioned perspectives, the subject of the present article is pertinent, and its findings may hold relevance for an audience of readers of Pathophysiology. Concomitantly, a series of observations have emerged during the review process of this scholarly work.
General comments.
The authors of the study hypothesize that Mpro may directly impact glucose transport and metabolism in HepG2 cells during the course of SARS-CoV-2 infection. Meanwhile, I believe that there are no sufficient grounds for this assumption. Firstly, in the event of an infection, Mpro functions within the confines of cells rather than from the extracellular environment, as was observed in the experimental setting. Secondly, the observed effect is attributable exclusively to concentrations of Mpro that induce partial cell death, thereby engendering stress in the surviving cells. The presence of cellular stress is confirmed by increased production of IL-6 and MMP1 by cells, as well as signs of oxidative stress. Concurrently, it is acknowledged that stress serine protein kinases have the capacity to impede insulin receptors, and a multitude of additional mechanisms of cellular pro-inflammatory stress have been identified, exerting influence on the expression and signaling pathways of insulin receptors.
More specific remarks.
(1) Abstract. "Levels of inflammatory markers, such as interleukin-6 (IL-6), were found to be elevated in the extracellular medium, while the expression of dipeptidyl peptidase-4 (DPP4) was found to be decreased." Conversely, the data presented in Figure 3.7 indicates that the observed alterations in DPP4 do not attain statistical significance.
(2) Introduction.   “…particularly in the development of insulin resistance and new-onset DM”. The preferred form is to write the full name of the disease for the first time: "type 2 diabetes mellitus." To ensure the study's efficacy, the objective should be articulated with greater precision.
(3) Results. The name of Figure 3.1. does not correspond to its content. The number of asterisks (*) on many Figures in the notes does not correspond to their number on the Figure itself. In the note in Figure 3.7, it is necessary to talk not about cells, but about the intercellular environment.
(4) DISCUSSION. The authors expound extensively on the role of the glucose transporter GLUT2 in hepatocytes. In contrast to GLUT4 in skeletal muscles and adipocytes, the effect of insulin in hepatocytes is primarily realized through signaling pathways rather than glucose transport: IRS/PI3K/AKT2/GSK3/glycogen synthesis and IRS/PI3K/AKT2/FOXO1/inhibition of gluconeogenesis. Concurrently, the two pathways are subject to inhibition by numerous pro-inflammatory cellular stress factors. Next, the authors associate the function of MMP1, with the action on extracellular matrix proteins. Meanwhile, we can talk more specifically about the effect of MMP1 on hepatocytes and other liver cells through the PAR1 receptor. I recommend that the authors supplement this section with the subsection limitation of research, in which you can point out the problems I outlined above.
(5) References should be adjusted to the MDPI style.
